# Biomarkers for Salvage Therapy in Testicular Germ Cell Tumors

**DOI:** 10.3390/ijms242316872

**Published:** 2023-11-28

**Authors:** Milena Urbini, Sara Bleve, Giuseppe Schepisi, Cecilia Menna, Giorgia Gurioli, Caterina Gianni, Ugo De Giorgi

**Affiliations:** 1Biosciences Laboratory, IRCCS Istituto Romagnolo per lo Studio dei Tumori (IRST) “Dino Amadori”, 47014 Meldola, Italy; giorgia.gurioli@irst.emr.it; 2Department of Medical Oncology, IRCCS Istituto Romagnolo per lo Studio dei Tumori (IRST) “Dino Amadori”, 47014 Meldola, Italy; sara.bleve@irst.emr.it (S.B.); giuseppe.schepisi@irst.emr.it (G.S.); cecilia.menna@irst.emr.it (C.M.); caterina.gianni@irst.emr.it (C.G.); ugo.degiorgi@irst.emr.it (U.D.G.)

**Keywords:** germ cell tumors, testicular cancers, salvage therapy, biomarkers, immunotherapy, molecular alterations, microRNA

## Abstract

The outcome of metastatic testicular germ cell tumor patients has been dramatically improved by cisplatin-based chemotherapy combinations. However, up to 30% of patients with advanced disease relapse after first-line therapy and require salvage regimens, which include treatments with conventional-dose chemotherapy or high-dose chemotherapy with autologous stem cell transplantation. For these patients, prognosis estimation represents an essential step in the choice of medical treatment but still remains a complex challenge. The available histological, clinical, and biochemical parameters attempt to define the prognosis, but they do not reflect the tumor’s molecular and pathological features and do not predict who will exhibit resistance to the several treatments. Molecular selection of patients and validated biomarkers are highly needed in order to improve current risk stratification and identify novel therapeutic approaches for patients with recurrent disease. Biomolecular biomarkers, including microRNAs, gene expression profiles, and immune-related biomarkers are currently under investigation in testicular germ cell tumors and could potentially hold a prominent place in the future treatment selection and prognostication of these tumors. The aim of this review is to summarize current scientific data regarding prognostic and predictive biomarkers for salvage therapy in testicular germ cell tumors.

## 1. Introduction

Estimating the prognosis of patients with metastatic testicular germ cell tumors who experience treatment failure with cisplatin-based first-line chemotherapy is a complex challenge. Currently, the treatment choice is guided by histological, clinical, and biochemical parameters that attempt to define the prognosis.

Markers that represent the tumor’s molecular and pathological features are still lacking and the prediction of treatment resistance remains an unmet need in everyday clinical practice.

In the context of refractory/relapsed metastatic GCTs, the identification and validation of molecular biomarkers could potentially improve current risk stratification and identify new therapeutic approaches as salvage therapies for these patients.

Several biomolecular biomarkers are currently under development in GCTs. Among these, microRNAs, gene expression profiling, and the presence of immune-related biomarkers could potentially play a role in prognosis, treatment selection, and stratification of patients with recurrent disease.

In this review, we summarize current scientific data regarding prognostic and predictive biomarkers for salvage therapy in testicular germ cell tumors, focusing on clinical, molecular alterations, microRNAs, and immuno-related markers.

## 2. Prognostic Model for Patients Who Failed Cisplatin-Based First-Line Chemotherapy

Germ cell tumors (GCTs) are the most prevalent tumors in young men, accounting for 1% of all male cancers and 5% of all male genitourinary malignancies [1,2,3].

Platinum-based chemotherapy has improved the prognosis of individuals with disseminated GCTs, with an 80% chance of complete remission. Although the combined therapies produce long-term results in the majority of patients, there is a subset for whom prognosis is less favorable [4].

Patients with poor prognosis features at the time of diagnosis, such as large tumor masses, multiple metastasis sites, extrapulmonary visceral metastases, or high levels of tumor markers, have a chance of cure of less than 50% with standard first-line therapy. Indeed, 20–30% of these patients do not achieve an initial complete response (CR) with first-line therapy and eventually relapse [5,6].

Salvage chemotherapy with cisplatin achieves long-term remission in up to 50% of seminoma patients with recurrence after first-line chemotherapy, and 20–50% of nonseminoma patients, depending on the presence or absence of specific risk factors. An incomplete response, high tumor volume and/or high levels of tumor markers, mediastinal primitiveness, presence of non-pulmonary visceral metastatic sites, and late relapses are all risk factors.

Four cycles of PEI/VIP (cisplatin, etoposide, ifosfamide), four cycles of VeIP (vinblastine, ifosfamide, cisplatin), or four cycles of TIP (paclitaxel, ifosfamide, cisplatin), or GIP (gemcitabine, ifosfamide, and cisplatin) are standard-dose chemotherapy regimens for recurrent disease [7,8,9].

High-dose chemotherapy (HDCT) with the assistance of peripheral blood progenitor cells (PBPCs) is another viable alternative for cisplatin-refractory GCT patients, particularly those with unfavorable features [10,11,12].

Prognostic variables at first diagnosis are widely recognized for patients with metastatic GCTs, and the International Germ-Cell Cancer Cooperative Group (IGCCCG) Classification is used to guide the treatment choice for first-line therapy of patients.

However, the question of prognostic variables is significantly more complicated in the case of recurrence [13,14,15].

There are no biomarkers available to guide treatment decisions in patients with GCTs who experience treatment failure with cisplatin-based first-line chemotherapy, and great efforts have been made to develop prognostic indicators available in everyday clinical practice.

Patients with advanced GCTs who progress during or relapse after cisplatin-based first-line treatment are a very diverse group and the International Prognostic Factor Study group (IPFSG) has tried to develop a prognostic model.

This score classifies patients into five prognostic groups based on six different clinical variables: primary site; first-line response; platinum-free interval; presence of bone, liver, or brain metastasis; level of tumor markers human chorionic gonadotropin (HCG) and alpha-fetoprotein (AFP) at baseline of salvage chemotherapy (Figure 1).

The IPFSG score is a valuable tool that can aid treatment decisions. However, due to the population’s great complexity as well as the high rates of treatment-related toxicity, the search for new prognostic variables is required. Despite their limited sensitivity and varying manifestations in the various subtypes, current guidelines propose that the therapeutic monitoring of these patients is mostly dependent on the assessment of conventional blood tumor markers [16].

## 3. Genetic Determinants

GCTs are classified into three types: type I (teratomas and yolk sac tumors) seen in neonates and children; type II (seminomas and non-seminomas), the most common type in young adults; and type III (spermatocytic tumors), which mostly affects men over the age of 50 [17].

Currently there is no biomarker able to properly predict treatment outcome in type II GCTs. Globally, nonseminomas are somewhat less sensitive to cisplatin-based therapy with respect to seminomas; however, the histology is not sufficient for patient stratification [14,15]. Several molecular aspects have been investigated in recent years, with the advent of next-generation sequencing approaches and methodological improvement of genetic and epigenetic analyses (Figure 2).

According to The Cancer Genome Atlas (TCGA), GCTs exhibit no significant mutational burden, with only 0.5 mutations per Mb [18]. Tumor mutational burden (TMB) has been shown to be slightly higher in platinum-resistant GCTs. About 10–20% of GCTs carry KIT- and/or RAS-activating mutations, which are more frequently found on seminomas [19,20]. Correcting for histology, no association was found between the presence of these mutations and clinical outcome [20]. Differently from other solid tumors, TP53 is retained in the majority of GCTs, being one of the reasons for the high sensibility of GCTs to cisplatin. However, a minority of cases show p53 pathway inactivation. Bagrodia et al. indicated that *TP53* and *MDM2* alterations can be involved in cisplatin resistance in GCTs [21]. In fact, refractory/relapsed TGCTs and primary mediastinal GCTs have been reported to be characterized by frequent *MDM2* amplification and *TP53* mutations: the first being mostly found in testicular tumors, whereas *TP53* mutations were primarily found in mediastinal GCTs [19,22,23,24]. Interestingly, most *TP53* alterations were found to affect the DNA-binding domain, a well-known mutational hotspot [22]. An update concerning molecular determinants in cisplatin resistance and organotropism in GCTs was presented by Bagrodia at ASCO 2023. Tumor mutational and transcriptomic profiles of a cohort of 138 patients with GCTs were analyzed with next-generation sequencing. In particular, primary and metastatic tumors seemed to have slightly different frequencies of putative driver gene mutations: *KIT* and *KRAS* were enriched in primary tumors while *TP53* was enriched in metastatic tissues. Interestingly, when only chemotherapy-naïve GCTs were considered, tumors having *MDM2* amplification tended to be the least sensitive to cisplatin, followed by *TP53*-mutant and *KRAS*-mutant tumors. However, these data are still preliminary and are not yet published in a peer-reviewed journal [25].

On the other side, copy number alterations are frequent, with TGCTs generally showing a hypertriploid genome with recurrent gain of chromosome 12p. Recently, a study conducted on cisplatin-resistant cell lines identified a copy number gain of chromosome 3 cytoband p25.3 as a possible driver of cisplatin resistance. Then it was evaluated in multiple cohorts of patients with type II GCTs (TGCA, MSKCC-2008, MSKCC-2016, MSKCC-2017), confirming that chromosome 3p25.3 gain was associated with cisplatin resistance, shorter progression-free survival, and overall survival. Moreover, this gain was associated with the presence of yolk sac histology and predicted adverse outcome independent of IGCCCG risk classification and the presence of *TP53*/*MDM2* alterations [26].

Interestingly, co-occurrence of *TP53* and 3p25.3 aberrations has been reported, in line with an enrichment in 3p25.3 gain in mediastinal tumors. However, p53 pathway inactivation and 3p25.3 gain seem to be two separate mechanisms leading to cisplatin resistance in GCTs, and cases with double alterations (*TP53* and chr3 gain) seem to have worse prognosis than tumors harboring only one alteration [26].

From a global point of view, COSMIC mutational signature 3 (associated with failure of DNA double-strand break repair by homologous recombination) have been found enriched in cisplatin-sensitive GCTs. Deregulation of the DNA repair pathway in cisplatin-resistant GCT cell lines was also demonstrated at the protein level [27]. Conversely, COSMIC signature 1 (associated with spontaneous deamination of 5-methylcytosine) is enriched in platinum resistance [19]. This is consistent with the indications that different methylation status could be implied with cisplatin resistance or sensitivity.

It is well known that, at the epigenetic level, TGCT histologies are different, with seminomas being usually unmethylated (or severely hypomethylated) tumors, embryonal carcinomas exhibiting low to intermediate methylation status, whereas teratomas and yolk sac tumors are hypermethylated [28]. Histology is the main determinant of the methylation level in TGCTs, however, changes of methylation status were found to be associated with platinum sensitivity. In particular, global hypermethylation, affecting both regulatory, intergenic, genic, and repeat elements, was associated with cisplatin resistance in isogenic cell line models and in matched primary and metastasis patient-derived tissues [29,30]. In more detail, Fazal found that hypermethylation was associated with CpG methylation of cancer suppressor genes and with nuclear organization of repressive chromatin (e.g., *CTCF*, *RAD21*, and lamina associated domains). These findings support a role of global nuclear reorganization of the heterochromatin structure in the platinum resistance process. Conversely, in platinum-sensitive cell lines, CpG islands associated with H3K27me3 and polycomb pathway were hypomethylated, in particular on EZH2 and SUZ12 binding sites [30]. These findings indicate that cisplatin resistance is associated with a global decrease in H3K27me3 and enrichment in polycomb target gene expression [30,31]. In an epigenetic study conducted by Lobo et al. on patient tumor samples, these findings were only partially confirmed, probably due to the intrinsic higher heterogeneity of patient-derived samples with respect to cell lines. Nonetheless, Lobo et al. confirmed the involvement of hypomethylated regions affecting chromatin remodeling and DNA binding. In addition, hypermethylation of promoters of genes involved in immune cell regulation was found in resistant TGCT tissue [29].

A different study highlighted the presence of a reciprocal epigenetic change between cisplatin and 5-aza (hypomethylating agent) in TGCT cell lines [31], supporting the presence of distinct epigenetic vulnerabilities that could be used as molecular targets for novel therapies. In fact, the authors found that cisplatin resistance increases sensitivity to 5-aza, and that the induction of 5-aza resistance increases sensitivity to cisplatin. At the transcriptional level, they confirmed the involvement of polycomb genes: in cisplatin-resistant cells, polycomb target genes were upregulated; conversely, in 5-aza resistance, they were downregulated. Moreover, increased H3K27me3 and decreased *DNMT3B* levels were found in 5-aza resistance, while the opposite was detected in cisplatin resistance [32].

At the transcriptional level, gene signatures associated with cisplatin resistance have been searched. In a study analyzing cancer testis genes (1036 testis-specific expressed protein-coding genes and 863 testis-specific expressed long noncoding RNAs) the authors demonstrated that cancer testis genes were more expressed in seminoma compared to nonseminoma histology. Moreover, unlike other malignancies, 96.16% of those genes were downregulated in GCT samples with respect to normal testis, while genes involved in stem cell maintenance-related pathways were upregulated. Interestingly, the expression levels of these last genes (e.g., *TSGA10*, *AKAP4*) were found to be associated with clinical outcome and could serve as prognostic factors [33].

Finally, a different mechanism proposed to be involved in cisplatin resistance is neddylation, a posttranslational modification process mediating many important biological processes, including tumorigenesis. Upregulation of *NAE1*, a key component of neddylation, has been identified using CRISPR/Cas9 activation screening on TGCT cell lines. The use of inhibitors of neddylation reverted cisplatin resistance back to sensitivity [34]. Other studies will be needed to fully address the role and crosstalk between all these determinants in the determination of cisplatin resistance.

## 4. MicroRNA Dysregulation

MicroRNAs (miRNAs) are a class of short non-coding RNAs that are involved in the epigenetic regulation of the expression of protein-coding genes binding the 3′UTRs of numerous mRNA transcripts. MiRNAs play critical roles in several physiological and pathological processes, including cancer. Dysregulation of miRNA expression is involved in the tumorigenesis process, acting as both oncogenes when overexpressed or as tumor suppressor genes when underexpressed [35].

MiRNAs are stable molecules present in high concentrations in blood that may be extracted non-invasively from urine or sperm and are easily identified by common molecular biology procedures such as PCR. Their utility as biomarkers has previously been proven in a variety of malignancies, including genitourinary cancers. In recent years, miRNAs have demonstrated their potential role as diagnostic and prognostic tools for GCT patients [36,37].

The use of blood levels of microRNAs (miRs) from the miR-371-3 and miR-302/367 clusters as potential GCT indicators was initially proposed in 2011 [38]. These two clusters of miRNAs were demonstrated to be highly expressed in GCT tissue and in the serum of GCT patients, while they were absent in other malignancies. Then, several other studies have shown the implication of several miRNAs in different aspects of TGCT biology, such as tumor initiation, progression, and drug resistance [39,40,41]. Among these, miR-372, miR-373, miR-34a, and miR-520c target genes are involved in cell cycle regulation and apoptosis, inhibiting tumor growth and promoting cell death in TGCT patients. Conversely, the miR-17-92 cluster, the members of which act as oncogenes, are overexpressed in TGCTs and promote cell proliferation and survival by targeting tumor suppressor genes [42,43,44,45].

Interestingly, the expression profiles of certain miRNAs in tumor tissues or circulating miRNAs in serum have been linked with clinical stage and tumor volume, with high sensitivity and specificity [44,45].

In particular, in a sizable cohort (n = 616) of patients affected by testicular GCTs at various stages, the study led by Dieckmann prospectively assessed the function of the M371 test, an assay specific for the detection of miR-371-3, for the monitoring of GCT treatment. Globally, the M371 test showed a sensitivity of 90.1% and specificity of 94.0% for the diagnosis of primary GCTs. Then, focusing on patients with recurrent disease, higher levels of miR-371-3 were discovered in 38 out of 46 patients, and it was found that the miRNA level decreased following treatment. Relapsed patients (n = 46) exhibited substantially higher median serum miR levels than controls (*p* = 0.001). Elevated levels were identified in 38 individuals, resulting in a sensitivity of 82.6%, specificity of 96.1%, and AUC of 0.921 for detecting relapse. Serial miR level assessments during therapy demonstrated substantial reductions in 28 of 29 patients [42]. The current consensus is that measures of miR-371a-3p outperform traditional markers, so clinical use of the test is justified.

Another relevant key issue in TGCT treatment is chemotherapy resistance. MiRNAs have been correlated with drug resistance via regulation of drug transporters and pro-survival pathway expression. Specific miRNA expression profiles were identified to be associated with drug resistance in GCTs. These profiles can potentially serve as predictive biomarkers of treatment response and help guide personalized therapeutic strategies. Several studies conducted on paired combinations of CDDP-sensitive and -resistant TGCT cell lines of diverse origins and histologies and miRNA expression patterns were evaluated in order to discover miRNAs possibly functioning as predictors of CDDP responsiveness in TGCTs. Regarding cisplatin resistance, several miRNAs were reported. In particular, the relevance of the miR-371-373 cluster was also confirmed in this setting [46], and additional novel potential miRNAs biomarkers associated with chemoresistance and aggressive phenotypes of TGCTs were identified (miR-512-3p, miR-515, miR-518, miR-525-3p, miR-218-5p, miR-31-5p, miR-375-5p, miR-517-3p, miR-20b-5p, and miR-378a-3p) [46,47]. On the contrary, other studies have highlighted the role of microRNAs in enhancing cisplatin sensitivity, in particular of miR-302a, miR-106b, and miR-383, which are involved in the downregulation of p21 and induction of cell cycle arrest [48,49,50].

The correlation between miRNAs and drug resistance in GCTs is a complex and multifaceted phenomenon. MiRNAs can influence drug resistance through various mechanisms, and their dysregulation can lead to altered drug response in GCTs. MiRNAs regulate drug transporters, apoptosis, DNA repair, CSCs, the EMT, and interactions with the tumor microenvironment, all of which contribute to treatment resistance in cancer patients [51,52]. Understanding the precise miRNA profiles linked with drug resistance in GCTs may contribute to the development of more effective targeted medicines and treatment methods for this difficult category of tumors. More study is needed to understand the complex regulatory networks involving miRNAs in TGCTs and to investigate their therapeutic potential, such as employing miRNA-based treatments or miRNA mimics and inhibitors. Understanding the involvement of miRNAs in TGCTs might eventually lead to better diagnosis, prognosis, and therapy options for this cancer.

## 5. PD-1/PDL-1/TIGIT/Inflammatory Biomarkers

The advent of the first immune checkpoint inhibitors led researchers to look for novel immunological biomarkers that could be considered as prognostic and predictive for immunotherapy efficacy against tumors [53]. Among them, the programmed-death receptor (PD-1) pathway was evaluated in GCT patients, with several studies demonstrating that programmed-death ligand 1 (PD-L1) and CTLA4 are frequently expressed in GCTs [54,55,56,57]. In the TCGA database, a “T-cell inflamed gene” signature was reported in 47% of GCT samples [55].

In particular, programmed-death receptor (PD-1) pathway markers were differentially expressed among GCT histologies. PD-L1 is frequently expressed in GCTs, at higher frequency in seminomas with respect to non seminomas [53,58,59,60]. In general, PD-L1-positive TILs were reported in 85.9% of seminomas, 91% of embryonal carcinomas, 54.5% of choriocarcinomas, 60% of yolk sac tumors, and 35.7% of teratomas [60]. Similarly, PD-L1-positive tumor-associated macrophages (TAMs) were found to be significantly expanded in seminomas compared with NSGCTs [59,60]. Conversely CTLA4 was expressed in 89.7% of GCTs, with higher intensity in yolk sac tumors, choriocarcinomas, and teratomas [56]. Interestingly, CTLA4 and PD-L1 expression was not significantly related to other GCT features, such as tumor stage, IGCCCG grouping, rete testis or lymphovascular invasion, nor between CTLA intensity and recurrence-free survival (RFS) [56].

Moreover, an inverse correlation between the T-cell–inflamed tumor microenvironment (TME) and AFP levels, especially in seminoma cases, was reported [55].

Despite low CD8+ T cell activity being recognized in GCT tissue [61], the prognostic role of PD-L1 expression on tumor infiltrating lymphocytes (TILs) was demonstrated in different studies [57,62,63]. In particular, Chovanec et al. demonstrated that a high infiltration level of PD-L1-positive TILs in TGCTs was associated with higher PFS and OS [63], while Boldrini et al. demonstrated that PD-L1 expression on TILs is a favorable prognostic factor in malignant extracranial GCTs and that a high density of T cells tended to affect the RFS of patients with gonadal GCTs [62]. In another study, utilizing a multiplicative quick score, lower PD-L1 expression significantly increased PFS (HR = 0.40; *p* = 0.008) and OS (HR = 0.43; *p* = 0.040) [57].

Notwithstanding, the predictive role of PD-L1 expression is still uncertain, not only in GCTs. In fact, several immune responses against PD-L1-negative tumors were often reported [64]. Perhaps more complete knowledge of antitumor immune machinery will explain this incongruence in the near future.

On the other hand, in a comprehensive molecular characterization, no significant neoantigen signals in GCTs were reported, suggesting that the low GCT mutational burden could explain the disappointing results of immune checkpoint inhibitors against this tumor type [18].

More recently, a correlation was reported among (1) lower TME expression of PD-L1 and immune checkpoint protein V-domain Ig suppressor of T cell activation (VISTA), (2) higher platelet-to-lymphocyte ratio, and (3) higher risk of events, probably because of the involvement of both local and systemic immune responses against GCTs [65].

The systemic immune-infiltration index (SII), a marker of proinflammatory TME, was also tested in GCTs. In one study, SII was tested together with other biomarkers, such as low albumin and hemoglobin, high leukocytes, neutrophils, C-reactive protein (CRP), and the neutrophil-to-lymphocyte ratio (NLR), with the authors demonstrating their correlation with poor prognosis in GCTs [66]. The correlation between higher SII levels and poor prognosis was reported in two independent GCT patient cohorts. In the same study, the authors reported both a prognostic inverse correlation between SII and PD-L1 expression on TILs; indeed, a better prognosis was reported in cases with low SII and high PD-L1 on TILs [66,67]. The same results were recently confirmed in two other trials [68,69].

Other immune checkpoints were evaluated in GCTs. The expression of T cell immunoreceptor with Ig and ITIM domains (TIGIT) was evaluated in 78 seminoma samples by Hinsch et al., who reported its frequent hyperexpression, often in combination with PD-1 positivity in T cells [70]. The same correlation was reported for higher PD-L1 and lymphocyte activation gene-3 (LAG-3) expression, the latter being involved in immune homeostasis by blocking T cell activation and cytokine secretion [71]. Instead, T cell immunoglobulin and mucin domain-3 (TIM-3) is involved in T cell exhaustion, and it could be involved in PD-1 monotherapy treatment failure [72,73]. Notwithstanding, neither LAG3 nor TIM3 expression levels in GCT cells were higher than in normal cells [74].

Recently, mismatch repair (MMR) deficiency was tested as a potentially related marker to PD-L1 expression in GCTs [54]. This condition correlates with immune-sensitive status, microsatellite instability (MSI), and platinum resistance in GCTs [75,76].

MiRNAs were also tested in GCTs. A study reported the significant expression of miR-125b, which is involved in the secretion of tumor-derived chemokines such as CSF1 and CX3CL1, which in turn is involved in TAM recruitment [77].

Several soluble molecules, such as IL-8, increase NF-kB and ABCB1, which in turn reduces cisplatin sensitivity in different tumor types, including GCTs [78]. It is noteworthy that platinum-derived cell destruction could be responsible for the secretion of several protumoral factors in the cancer stroma [79]. Another study demonstrated the prognostic role of proinflammatory cytokines, such as IFN-a2, IL-2Ra, or IL-16, in GCTs [80]. Other studies are evaluating different molecules, such as: (1) IL13RA2, which is hyperexpressed in normal testis cells and is being studied as a potential CAR target against several neoplasms [81]; and (2) b1,4-galactosyltransferase-I (B4GALT1), an enzyme involved in interaction and adhesion of immune cells; for which higher expression in peripheral T cells was correlated with a lower risk of GCT relapse after salvage high-dose chemotherapy by Nilius et al. These findings were probably due to B4GALT1-mediated activation of CD4+ T cells, which was correlated with IL-10 hyperexpression and, consequently, better prognosis in GCTs [82].

Among novel biomarkers, great interest in nanoparticles in the research community is due to the fact that they are the base for the construction of multifunctional nanoscale devices, which also contribute to GCT biomarkers and drug delivery [83,84].

Finally, CAR-T cell therapy is currently under investigation in GCTs. CAR-T cell therapy consists of genetically engineered T cells expressing antigen-specific receptors on their cell membranes. This structure is composed of four regions: an antigen-binding domain; a hinge region, which connects scFV with the transmembrane region; and an intracellular extremity, which comprises the signal transduction part of the TCR linked with one or two costimulatory domains [85]. The advantages of CAR-T cell therapy are as follows: (1) the immune mechanism of action is not MHC restricted [86], and (2) its low antigen affinity in TCRs can induce off-target toxicities [87]. Moreover, the T cell lytic property represents another CAR-T cell characteristic [88].

CAR-T cell therapy has demonstrated its efficacy in hematological tumors, but currently this is not the case for solid neoplasms due to their intra-tumor heterogeneity and TME activity [89]. Notwithstanding the above, novel early-phase trials presented at ESMO Congress 2022 could be a step forward in CAR-T cell therapy development for non-hematological neoplasms. Among them, a phase I first-in-human trial tested BNT211, an autologous CAR-T cell therapy targeting the oncofetal antigen claudin-6 (CLDN6), and a CLDN6-encoding CAR-T cell amplifying RNA vaccine (CARVac) in patients with CLDN6-positive relapsed/refractory non-hematological cancers, including 13 GCT patients [90]. CAR-T cell therapy was administered at two different dose levels. In the GCT patient cohort, an ORR of 57% and DCR of 85% (1 CR, 3 PR, 2 SD) were reported. CAR-T cell therapy was well tolerated, but pancytopenia in the monotherapy cohort and hemophagocytic lymphohistiocytosis in the combination cohort were reported. The results of the study are still limited and premature, but they provide an excellent starting point for further investigations.

## 6. Conclusions

Long-term remission can be achieved by salvage therapies in patients with recurrence after first-line chemotherapy. However, a significant fraction of patients still experience relapse or incomplete response. Several biomarkers have been studied in recent years, covering clinical to molecular aspects. The IPFSG score is surely a valuable tool able to aid in treatment decisions, while on the other side, genomic alterations and the miRNA profile have shown promising results for patient stratification. Progressive deepening of the knowledge of the molecular and immunological features of refractory/resistant GCTs will help in defining alternative therapeutic strategies for the management of this disease.

## Figures and Tables

**Figure 1 ijms-24-16872-f001:**
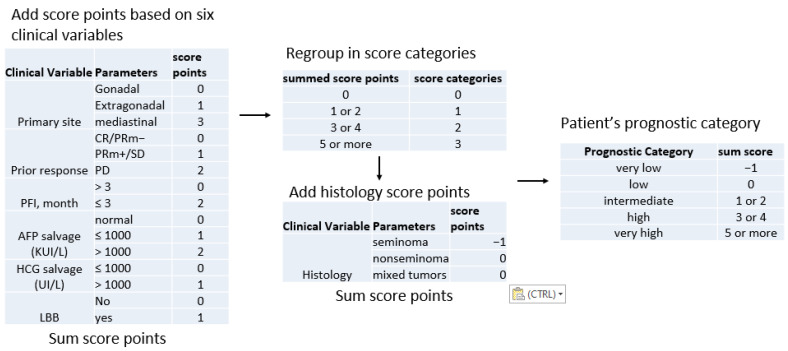
Workflow for IPFSG score determination. Abbreviations: CR, complete remission; PRm, partial remission with negative markers; PRm, partial remission with positive markers; SD, stable disease; PD, progressive disease; PFI, progression-free interval; AFP, alpha-fetoprotein; HCG, human chorionic gonadotrophin; LBB, liver, bone and brain metastases.

**Figure 2 ijms-24-16872-f002:**
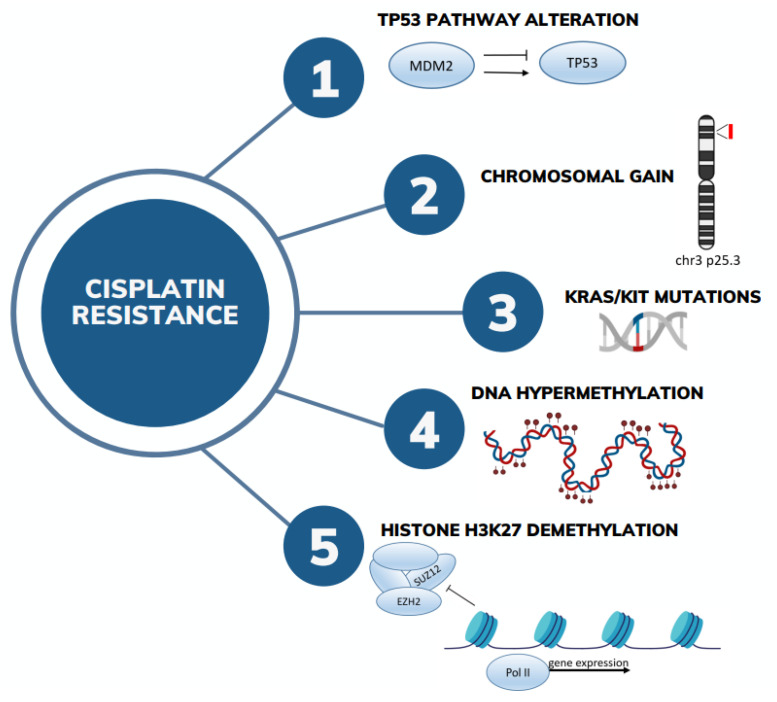
The most relevant molecular aspects involved in cisplatin resistance in GCTs.

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
