# Peer review of "Biomarkers for Salvage Therapy in Testicular Germ Cell Tumors"

_ijms, 2023, doi:10.3390/ijms242316872_

Round 1

Reviewer 1 Report

Comments and Suggestions for Authors

1) General comments

Urbini et al. reviewed and summarized current data on prognostic and predictive biomarkers for salvage therapy in testicular germ cell tumors (TGCT), including prognostic model, genetic aberrations, microRNA dysregulation, and immune status. This manuscript is generally well written, and I have a few comments for clarifying some aspects as below.

2) Specific comments

1. Figure 1: Some abbreviations, such as LBB (possibly Liver, Bone, and Brain?), needs to be explained in Figure legend. Unit for AFP and HCG should be added to the Figure.

2. Lines 206-210: This sentence starting “Coversely, ..” seems to be unrelated to the previous sentence. Please clearly state the relationship between these studies (refs 24 and 32).

3. Lines 298-301: Results of refs. 53-55, and their association are unclear. Detailed description is necessary.

4. Several abbreviations in the manuscript should be firstly spelled out: for example, line 74 HDCT, line 145 CAN, line 307 TME, and line 326 NLR.

Comments on the Quality of English Language

There are many typos in the manuscript, such as line 56 “GTCs” (seems GCTs), line 108 “spermatocyte tumors” (seems “spermatocytic”), line 140 “PD1L” (seems PD-L1).   Please carefully re-check the whole manuscript.

Author Response

Thanks for revising the manuscript. Answers are in the attachment.

Reviewer 2 Report

Comments and Suggestions for Authors

Urbini et. al . give a nice overview of the biomarkers for salvage therapy in testicular germ cell tumors. I it well written and generally well structured. The cited references are relevant and the overview of the literature is comprehensive, with no clear omissions of relevant recent literature. 

I do have a number of small comments. 

1. The section on immune biomarkers is not very well organized. It reads as an enumeration of a lot of correlative studies, with not much data supporting the clinical relevance. In my view it could be shorter with a bit more emphasis on the preliminary character of these studies. I also feel that CAR-T biomarkers does not warrant it's own subheading and could be combined with the previous chapter. 

2. At p7 l256-276 the authors talk about the putative role of miRNAs in drug resistance in GCTs. This is of course potentially very interesting, but there is only one GCT related publication mentioned with does not show very pervasive proof of this principle. I think this section should be shorter and should contain a clearer caveat. 

3. The ASCO meeting abstract of Bagrodia et al. is extensively discussed. I agree that is is very interesting, but it is not published and peer-reviewed yet.

4. p3l106-115 warrants some literature references. 

5. The introduction is the same as the abstract. 

Comments on the Quality of English Language

-

Author Response

Thank you for reviewing the manuscript. see the answers attached
